# Linguistic Markers in Spontaneous Speech: Insights into Subjective Cognitive Decline (Review)

**DOI:** 10.3390/healthcare13222888

**Published:** 2025-11-13

**Authors:** Sofia Segkouli, Mara Gkioka, Stylianos Kokkas, Konstantinos Votis, Sergi Valero, Andrea Miguel, Athos Antoniades, Emily Charalambous, George Manias

**Affiliations:** 1Centre for Research and Technology Hellas, Information Technologies Institute (CERTH/ITI), 57001 Thessaloniki, Greece; s.kokkas@iti.gr (S.K.); kvotis@iti.gr (K.V.); 2Institute of Applied Biosciences, Centre for Research and Technology Hellas (INAB|CERTH), 57001 Thessaloniki, Greece; mgkioka@certh.gr; 3Ace Alzheimer Center Barcelona, Universitat Internacional de Catalunya, 08029 Barcelona, Spain; svalero@fundacioace.org (S.V.); amiguel@fundacioace.org (A.M.); 4Networking Research Center on Neurodegenerative Diseases (CIBERNED), Instituto de Salud Carlos III, 28029 Madrid, Spain; 5Stremble Ventures Ltd., Limassol 4042, Cyprus; athos.antoniades@stremble.com (A.A.); emily.charalambous@stremble.com (E.C.); 6Department of Digital Systems, University of Piraeus, 18534 Piraeus, Greece; gmanias@unipi.gr

**Keywords:** population growth, Subjective Memory Decline (SMD), language impairment, spontaneous speech, linguistic changes

## Abstract

**Background and Objectives:** Population rapid growth and demographic shift is leading to a rise in neurodegenerative disorders such as dementia and mild cognitive impairment (MCI). Evidence indicates that MCI is not the earliest phase of prodromal AD. Subjective Memory Decline (SMD) refers to a self-perceived decline in cognitive abilities compared to previous functioning levels in individuals with normal cognition. Language impairment represents a critical marker of neurodegenerative disorders and early memory decline in healthy older adults. **Methods:** This review was conducted in accordance with PRISMA Statement guidelines. The inclusion criteria of the selection process were set as follows: (1) All studies analyzed spontaneous speech samples in individuals with SMD or individuals with +αβ amyloid. (2) Studies reported language performance indicators (e.g., lexical, syntactic, semantic, phonetic, or fluency measures) derived from spontaneous speech. (3) The study population included participants with SMD based on recognized diagnostic criteria or self-reported cognitive complaints without objective cognitive impairment. (4) Studies were written in English. (5) The time frame of studies was 5 years. **Results:** The present work is a review of speech features—particularly from spontaneous and narrative speech—and methods that can serve as sensitive indicators of early cognitive changes due to AD pathology. **Conclusions:** Spontaneous speech analysis, through acoustic and temporal parameters such as silence duration, phrasal segment length, and speech segment frequency, offers a rich window into the subtle cognitive and linguistic changes that reflect early memory decline in healthy older adults. Spontaneous speech performance could be a scalable, low-cost, and non-invasive diagnostic tool in proactive cognitive health.

## 1. Introduction

Population rapid growth and demographic shift is leading to a rise in neurodegenerative disorders such as dementia and mild cognitive impairment (MCI) [1]. MCI represents a transitional phase between normal aging and dementia, where individuals retain most functional abilities despite exhibiting neuropsychological impairments [2], and is considered a significant risk factor for Alzheimer’s disease (AD) [3]. Extensive research has explored MCI as a predictor of AD, including the development of psychological assessments targeting cognitive domains affected by MCI and dementia to distinguish healthy individuals from those exhibiting clinical symptoms [4]. 

However, evidence indicates that MCI is not the earliest phase of prodromal AD, highlighting the need for earlier disease detection to enable more effective interventions [5,6]. Some individuals report experiencing subtle changes in memory and cognitive function before any detectable cognitive impairment. Subjective Memory Decline (SMD) refers to a self-perceived decline in cognitive abilities compared to previous functioning levels in individuals with normal cognition [7,8] and has been recognized as a key factor in identifying prodromal AD [9]. The association between SMD and actual memory function remains unclear. Thus, those experiencing SMD report subjective memory issues or changes that are too minor to be detected by standard cognitive assessments [10]. However, individuals with these concerns tend to exhibit greater objective memory decline, lower overall cognitive performance, and poorer perceived health compared to those without SMD [11]. In the literature, SMD is also referred to as subjective cognitive decline (SCD), subjective memory impairment (SMI), and subjective cognitive complaint (SCC). While these terms vary slightly in definition, they all generally describe an individual’s subjective perception of their cognitive abilities.

Although there are no globally accepted or widely used criteria for diagnosing SCD, according Jessen and colleagues (2014), the SCD-I Working Group has introduced the SCD-plus criteria as a guideline [6]. These criteria incorporate biomarkers along with specific characteristics, such as self-perceived memory decline and the perception of having worse memory performance compared to peers of the same age. According to the SCD-I Working Group, the diagnostic criteria for SCD include (a) a subjective sense of worsening memory that is not linked to depressive symptoms, (b) the absence of objective cognitive deficits based on neuropsychological assessments, and (c) classification at stage 2 of the disease as defined by the Global Deterioration Scale (CDR) [12].

### 1.1. Spontaneous and Connected Speech as Markers of Detection in Cognitive Decline

Language impairment represents a critical marker of neurodegenerative disorders; however, the absence of a standardized terminology system for characterizing these impairments contributes to substantial inter-rater variability among clinicians. Recently, the application of natural language processing (NLP) techniques and automated speech analysis (ASA) has emerged as a novel and potentially more objective approach for evaluating language function in individuals with mild cognitive impairment (MCI) and Alzheimer’s disease (AD) [13].

Spontaneous speech is speech produced with minimal premeditation, constitutes effortless, natural, content-driven, and meaningful communication, and is used to describe unscripted, real-life verbal production reflecting cognitive–linguistic processes [14]. Studies have shown that individuals with high amyloid burden, a hallmark of Alzheimer’s pathology, exhibit a reduction in the use of specific words during spontaneous speech tasks. Other studies have focused on word count in semantic fluency or picture description tasks for classification purposes among MCI, SMD, and healthy older adults [15,16]. Moreover, machine learning techniques applied to paralinguistic features from brief spontaneous speech protocols have proven effective in distinguishing between varying degrees of cognitive impairment, including SCD, MCI, and AD [17]. These findings underscore the potential of speech analysis as a non-invasive, cost-effective method for early detection and monitoring of cognitive decline.

The analysis of connected speech has gained increasing importance over the past two decades, particularly in AD research, due to its ability to simultaneously engage multiple cognitive processes such as semantic storage and retrieval, executive functions, and working memory.

Connected speech is a term used to refer to any spoken language that flows continuously, not just isolated words, including all the sound changes that happen when one word connects to another in spontaneous speech [18].

Specifically, it refers to continuous sequences of spoken language where words flow naturally and phonetic/phonological processes (like assimilation, elision, and liaison) occur across word boundaries. Compared to isolated tasks like picture naming, connected speech offers a more comprehensive and ecologically valid assessment with minimal participant burden [19]. Although most studies have focused on individuals with mild to moderate AD, retrospective analyses [20,21] have identified linguistic changes as early as the MCI stage. Nevertheless, detecting subtle, preclinical language impairments remains challenging, partly due to the strong interdependence of language and memory functions and the lack of standardized methods to differentiate between them. Furthermore, distinguishing language changes in MCI from those related to normal aging or SCD is difficult [19]. Simultaneously quantifying both memory and language impairments may enhance diagnostic sensitivity and provide valuable prognostic information.

### 1.2. Speech Domains Connected to AD, MCI, and SMD

While episodic memory impairment or memory loss is a defining feature of AD, patients also develop key language deficits as a consequence of multiple cognitive impairments due to focal brain damage [22]. These can manifest as a decline in lexical semantic knowledge and difficulties in finding the right words (i.e., anomia and semantic paraphasias), and following a conversation, their fluency and rate of speech (phonetic level) are reduced while using grammatically correct but content-poor sentences [23,24,25,26,27]. Based on a number of studies, morphosyntactic processing remains mostly intact at first; nonetheless, individuals with AD make significantly more inflectional errors (mistakes with prefixes and suffixes that change a word’s form) than healthy older adults [25,28,29,30,31]. Critically, at the pragmatic level, their ability to hold a cohesive conversation also suffers, impacting how they link ideas (referential/temporal cohesion), maintain clarity, and organize their speech [32,33,34,35].

Verbal deficits observed in MCI mirror those of early-/moderate-stage dementia [36], specifically involving naming, verbal fluency, and semantic knowledge, where pragmatic skills seem to be the most affected area. It is well documented that these discourse issues are often the earliest detectable signs of AD pathology, sometimes appearing years before other memory or cognitive problems are formally diagnosed [37,38]. Therefore, analyzing these language markers is promising for both early detection and large-scale screening for dementia.

Studies regarding SMD deficits in language domains are sparse. However, some recent studies indicate that word-finding patterns, especially those related to specific and concrete words, may be sensitive indicators for early cognitive changes such as SMD [39,40,41]. The association with amyloid burden in SMD populations further supports their potential as early biomarkers of Alzheimer’s disease pathology. Specifically, Verfaillie et al. (2019) found that less use of words referring to human characteristics and concrete nouns, as well a lower use of content words, was associated with higher amyloid burden [40]. This indicates that a reduction in the density of meaningful words in spontaneous speech may reflect underlying neuropathological changes. In a longitudinal study (5-year study follow-up), Maruta and Martins (2019) provide insight into word retrieval efficiency [39]. While not directly measuring word-finding difficulties, poorer performance on semantic fluency tasks over time may reflect increasing challenges in lexical access. Another longitudinal study (5-year follow-up) by Reeves et al. (2023) indicates that a reduced word count combined with an increased frequency of interjections, or a lower narrative discourse (ND) score, was a significant predictor of subsequent cognitive decline [41].

### 1.3. Linguistic Metrics in Connected and Spontaneous Speech 

Current traditional methods of linguistic skills’ evaluation, typically involving paper-and-pencil or computer-based tasks that measure verbal fluency, visual confrontation naming, comprehension, and writing skills, remain limited. These conventional tests often lack the sensitivity required for early diagnosis and disregard more complex language dimensions such as prosody and speech rhythm (suprasegmental features). As a result, even when these standardized tools detect minor differences between participants with MCI and healthy older adults, their overall clinical utility is questionable and unreliable [36,42,43]. In recent years, the application of sophisticated natural language processing (NLP) techniques has revolutionized the analysis of language skills. By processing written texts, structured clinical speech, and spontaneous conversation, detailed linguistic features can be automatically extracted to help identify, classify, and describe signs of various psychiatric and neurological disorders. These computational methods have already proven successful in detecting the subtle linguistic markers that signal the very early stages of dementia [44,45,46,47] and characterizing associated conditions like AD [25,48,49,50].

With regard to metrics, research has indicated that “idea density” and “grammatical complexity” serve as indicators of dementia risk, with lower baseline levels of these measures linked to the development of dementia [51]. “Idea density”, or “propositional idea density (P-density)”, refers to the proportion of semantic content words relative to the total number of words in a sentence [52]. Other linguistic metrics used in connected language studies to assess semantic content include the ratio of nouns to the total number of pronouns and nouns, which quantifies “nonspecific language” by identifying pronouns without clear referents [53]. Additionally, connected language analysis has been used to measure grammatical complexity through verb percentage or verb indices [54] and to assess coherence and informativeness. A feature taxonomy has been described by de la Fuente Garcia (2020) [55] with regard to the metrics that have been defined per layer in spontaneous speech production: lexical features, syntactical features, semantic features, pragmatic features, prosodic features, spectral features, vocal quality, and ASR-related features. 

### 1.4. Tools to Measure Connected and Spontaneous Speech

Language samples have also been collected through more structured elicitation methods, such as open-ended questions or semi-structured interviews. For example, participants have been asked to describe “the happiest moment of their lives” [56] or to respond to general questions regarding their career, family, life experiences, and hobbies [57,58] in order to measure spontaneous speech. Other studies of connected speech have utilized more constrained tasks, such as picture description. While open-ended methods yield greater linguistic output, they tend to be highly variable across individuals and contexts, making standardization for comparative analysis challenging. In contrast, picture description tasks offer a structured means of assessment with standardized evaluation measures, and when the picture remains visible, they place less demand on episodic memory. The most frequently used picture stimuli in the literature include Norman Rockwell prints, such as “Easter Morning” [59], and the widely recognized “Cookie Theft” picture from the Boston Diagnostic Aphasia Examination [60]. The “Cookie Theft” picture is particularly notable, as it was designed to encompass elements of a person, time, place, and action while incorporating vocabulary that is typically acquired early in life [61].

Recent studies have employed various semi-spontaneous speech tasks. For example, one study protocol used three different prompts such as describing a complex image, detailing a typical working day, and recounting the last dream remembered [62].

Although there is extensive research on picture description in AD, studies focusing on this method in mild cognitive impairment (MCI) remain comparatively limited. However, smaller retrospective studies suggest that language decline may emerge in prodromal phases of the disease [21,38]. Picture description tasks could be valuable in detecting linguistic changes at the MCI or pre-MCI stage, like the SMD group, aiding in early diagnosis and identifying the optimal time to introduce cognitive–communication interventions.

### 1.5. Aim of the Review

The purpose of this literature review is to examine methods or tools for assessing language performance in individuals with SMD. Specifically, it aims to explore quantitative metrics and indicators of spontaneous speech that may serve as early markers of cognitive decline. By identifying such speech-based indicators, this review seeks to inform early diagnosis and help determine the optimal timing for initiating cognitive–communication interventions for individuals at risk of AD.

Research Questions:What characteristics of spontaneous speech have been used to detect cognitive decline in individuals with SMD?What linguistic metrics are used (e.g., phonological, syntactic, semantic, pragmatic, etc.)?What tools or methods are employed to analyze spontaneous and connected speech.

## 2. Materials and Methods

### 2.1. Search Strategy

A systematic literature search was conducted using two widely recognized databases: PubMed and Cochrane. The search employed a combination of keywords, subject headings, and MESH terms related to themes of Subjective Memory Decline and Spontaneous Speech (detailed in the Appendix A). The initial systematic search took place in March 2025.

### 2.2. Procedure, Selection Process, and Data Extraction

This review was conducted in accordance with PRISMA Statement guidelines [63]. Following the removal of duplicates, two reviewers (A.N. and M.G) independently screened articles based on their title, abstract, and relevance. Full-text screening was then performed for studies deemed potentially eligible. A third reviewer (MG) was involved to resolve any discrepancies between the initial two reviewers at each stage of the selection process. The inclusion criteria of the selection process were set as follows: (1) All studies analyzed spontaneous speech samples in individuals with SMD or individuals with +αβ amyloid. (2) Studies reported language performance indicators (e.g., lexical, syntactic, semantic, phonetic, or fluency measures) derived from spontaneous speech. (3) The study population included participants with SMD based on recognized diagnostic criteria or self-reported cognitive complaints without objective cognitive impairment. (4) Studies were written in English. (5) The time frame of studies was 5 years. The exclusion criteria were set as follows: (1) Studies focusing primarily on language markers in populations with MCI, AD, or other neurodegenerative conditions, without a distinct SMD or +αβ amyloid group. (2) Studies that assessed language performance using only structured or experimental tasks (e.g., naming tests, sentence repetition) without spontaneous speech analysis. (3) Case studies, reviews, opinion papers, reports, protocols, or meta-analyses. (4) Studies examining speech and language in psychiatric or neurological conditions unrelated to cognitive decline (e.g., aphasia, schizophrenia, stroke).

Any disagreements regarding eligibility criteria were resolved by a third reviewer (MG). Data extraction was performed independently by MG and SS using a standardized template that captured the following elements: 1. study/year; 2. methodology; 3. tools; 4. outcome measured; and 5. main findings.

### 2.3. Data Synthesis Strategy

The synthesis was carried out according to the methodology, the setting/participants, the tools used, the outcomes measured, and the main findings of the selected studies. These dimensions were used to interpret the findings and draw conclusions about language indicators in SCD.

## 3. Results

The review of the studies selected regarding the target group of SCD provided a full list of details according to the methodology, tools used, outcomes, and main findings, as summarized in Table 1.

Most of the studies had a cross-sectional design [17,40,55,64,65,67,68,69,70], except for van den Berg (2025) [66] and Reeves (2023) [41], which were longitudinal; Ter Huurne (2023) [71], which was a prospective cohort study; and Hajjar (2023) [72], which was a cross-sectional study with a 2-year follow-up. 

### 3.1. Articulatory and Prosodic Markers in Spontaneous Speech

The findings of the current review indicate that articulatory and prosodic features of spontaneous speech may serve as early indicators of cognitive decline in individuals with SMD. Specifically, slower speech rate, increased between-utterance pause time, and more frequent between-utterance pauses during delayed recall tasks have been significantly associated with elevated tau deposition in medial temporal and early neocortical regions, even in cognitively unimpaired adults [64]. These speech-based markers appeared to be independent of amyloid status and traditional delayed recall performance, suggesting their unique contribution to the early detection of Alzheimer’s pathology [64].

Furthermore, the phoneme log-likelihood ratio (PLLR), a computational measure of articulatory precision, demonstrated robust discriminatory power between cognitively unimpaired individuals with low amyloid-β burden and those diagnosed with mild cognitive impairment (MCI) or dementia [65]. The sensitivity of the PLLR to subtle differences between groups with milder cognitive deficits supports its potential utility as an early speech-based biomarker of neurodegeneration [65]. Similarly, a reduction in speech fluency and pace has been observed in individuals with MCI and dementia relative to cognitively unimpaired controls, providing further support for the relevance of temporal and articulatory markers in characterizing preclinical stages of cognitive impairment [65].

### 3.2. Lexical–Semantic and Language Content Features

Changes in the lexical and semantic properties of spontaneous speech have also emerged as potential indicators of early cognitive changes. Amyloid-positive individuals were found to produce fewer specific content words, particularly among highly educated participants, despite showing no significant alterations in lexical or syntactic complexity on conventional neuropsychological assessments [40]. This suggests that spontaneous speech tasks may offer enhanced sensitivity to detect subtle semantic impairments in preclinical Alzheimer’s disease. Declines in semantic verbal fluency have been consistently associated with an increased risk of clinical progression from SMD to MCI or dementia. Notably, semantic fluency declined more rapidly in amyloid-positive individuals compared to their amyloid-negative peers, while phonemic fluency remained stable [66]. Furthermore, a growing discrepancy between semantic and phonemic fluency scores over time has been linked with subsequent progression to MCI or Alzheimer’s dementia, indicating the clinical relevance of fluency metrics as longitudinal markers of risk [67].

The diagnostic utility of lexical–semantic and acoustic digital voice biomarkers was also supported by evidence showing that these features outperformed traditional neuropsychological tests in differentiating MCI from SCD [72]. Lexical–semantic indicators were able to detect amyloid-β status, while acoustic markers were correlated with hippocampal volume, reinforcing their biological validity [72]. Additionally, machine learning models analyzing paralinguistic features from spontaneous speech demonstrated high accuracy in identifying individuals with MCI and Alzheimer’s disease and predicting performance in multiple cognitive domains such as memory, executive function, and visuospatial ability [17].

### 3.3. Automated Speech Analysis and Remote Assessment Tools

Evidence from the current review supports the feasibility and diagnostic relevance of remote and automated speech assessments for individuals with SMD. Studies have shown that automatic speech recognition (ASR) technologies exhibit high agreement with manual transcription for word count metrics, with a 93% probability that discrepancies remain within minimally important differences [69]. However, qualitative features such as semantic cluster size and word frequency showed only fair levels of agreement between automated and manual transcription, indicating current technological limitations in more complex linguistic analyses [69]. Notably, the diagnostic accuracy of automated speech features has been demonstrated to differentiate between cognitively unimpaired, SMD, MCI, and AD groups, particularly when analyzing spontaneous speech and verbal fluency tasks using mobile applications [70]. The application of these tools is further validated in studies showing the high classification accuracy and predictive capability of machine learning models that utilize ASR-derived features from verbal learning and recall tasks [71].

Importantly, the performance of ASR systems is influenced by the clinical status of participants. Transcripts of speech from cognitively healthy individuals had higher recognition confidence and lower error rates than those from individuals with SMD, MCI, or AD. Manual correction of transcripts improved classification performance for spontaneous speech but had limited effect on reading tasks [68]. Furthermore, the PROSPECT-AD study highlighted the utility of remote speech-based neurocognitive assessments in identifying early Alzheimer’s biomarkers. The study emphasized the capacity of such speech features to correlate with CSF biomarkers such as amyloid-β1-42 and phosphorylated tau, thereby linking remote speech analysis with established biological measures [69]. High adherence and usability scores in remote assessments further support the scalability and acceptability of these tools in preclinical detection settings [67].

In addition, remote protocols, including phone-based and app-based speech evaluations, have shown high adherence and user satisfaction among participants, even in preclinical stages of AD [67]. In particular, tasks such as picture description and narrative storytelling were successfully administered remotely and demonstrated sensitivity to differences in amyloid status [67]. These findings support the integration of speech-based tools into scalable, accessible cognitive monitoring frameworks, especially for individuals reporting subjective cognitive complaints.

### 3.4. Speech vs. Traditional Neuropsychological Testing

Spontaneous speech analysis may offer distinct advantages over conventional neuropsychological assessments, particularly in the context of early or subtle cognitive deficits. Several studies have shown that linguistic and paralinguistic features extracted from natural speech can predict disease progression, outperform traditional test scores, and provide insight into multiple cognitive domains [17,72].

Although automated and speech-based tools may currently be limited in their ability to capture nuanced qualitative language features with the same reliability as manual assessments [69], their advantages in ecological validity, accessibility, and cost-effectiveness are considerable. Furthermore, these tools allow for continuous and remote monitoring, which is difficult to achieve with conventional testing.

However, challenges remain, as adding linguistic features to standard narrative description tests did not significantly improve predictive accuracy for future cognitive decline, suggesting that standalone speech analysis may not always yield additional diagnostic benefit [41]. Yet, as machine learning methods advance and data quality improves, the predictive and diagnostic utility of speech biomarkers is likely to grow, reinforcing their role alongside traditional cognitive assessments.

## 4. Discussion

The present review findings indicate that speech features—particularly from spontaneous and narrative speech—can serve as sensitive indicators of early cognitive changes due to AD pathology. According to current results, linguistic alterations, particularly in spontaneous and narrative speech, such as reduced fluency, slower speech rate, increased pauses, and decreased articulatory precision have been linked to biomarkers like amyloid and tau, even among asymptomatic individuals like SMD [40,64,65]. These subtle linguistic changes often emerge before overt clinical symptoms and are detectable through detailed acoustic and linguistic analysis. Automated speech analysis has emerged as a promising non-invasive tool for remote screening, early detection, and longitudinal monitoring of cognitive decline. In addition, with regard to our results, natural language processing (NLP) and machine learning algorithms enabled the identification of speech patterns that distinguish normal aging from MCI and various stages of cognitive decline with high levels of accuracy and specificity [17,70,72]. These technologies offer scalable, cost-effective, and user-friendly solutions, especially valuable in under-resourced settings or for populations with limited access to traditional healthcare services. In contrast, conventional neuropsychological assessments appear less sensitive to these early linguistic changes, highlighting the added value of speech-derived biomarkers in clinical evaluations [40,41,71]. The present study leverages the potential that the performance of spontaneous speech has for the identification and very early diagnosis of cognitive alterations in older adults. Moreover, beyond diagnosis through conventional neuropsychological tests of language, the design and implementation of tasks that elicit spontaneous speech in real contexts have proven beneficial. 

The findings are largely consistent with previous studies, reinforcing the growing consensus that speech features—particularly those derived from spontaneous and narrative speech—serve as sensitive early markers of cognitive decline. Our results align with similar studies [73,74] showing that subtle alterations in fluency, articulation, and speech timing can be detected even in asymptomatic individuals, such as those with SMD. Notably, automated speech analysis methods in our included studies, involved NLP and machine learning approaches, demonstrated high classification accuracy and predictive validity, similar to the AUC values (0.69–0.92) and accuracy rates (0.70–0.92) reported by similar studies such as Beltrami et al. (2018) [73] and Eyigoz et al. (2020) [75]. These findings further support the idea that speech-derived biomarkers outperform conventional language assessments like the Boston Naming Test [76], especially in identifying early, subtle cognitive changes that precede clinical impairment.

However, our review expands on the previous literature by emphasizing not only the diagnostic value of spontaneous speech but also the importance of ecologically valid, real-context elicitation tasks for capturing linguistic performance in its natural form. While similar studies [74,77] have acknowledged the role of discourse coherence, lexical retrieval, and sentence comprehension as early indicators, our included studies’ findings highlight the added value of context-sensitive speech tasks that may better reflect day-to-day communicative challenges in older adults. Furthermore, consistent with prior work [78], we observed that digital tools not only enhance diagnostic sensitivity but also offer scalable, low-cost, and user-friendly solutions—a particularly important advantage for under-resourced settings. 

Despite promising findings, several limitations must be addressed in the current review. A key limitation across the included studies lies in their considerable methodological heterogeneity, which complicates direct comparisons and the generalizability of findings. Variations in speech tasks (e.g., story recall, picture description, verbal fluency), speech processing tools (e.g., ELAN, Praat, WhisperX, OpenSMILE), and linguistic/acoustic features selected for analysis result in inconsistent outcome measures and interpretations [17,64,65]. Sample characteristics also differ notably across studies with varying diagnostic criteria and biomarker assessment methods [40,66] (PET vs. CSF) for amyloid. Automated systems likewise face challenges: although ASR achieves high agreement with manual transcription for basic metrics like word count, more nuanced features (semantic cluster size, word frequency, fluency patterns) only show fair concordance [69], potentially undermining clinical reliability. Furthermore, while some studies apply rigorous manual transcription and correction [68], others rely on automated methods that may introduce transcription errors affecting downstream analyses. Many findings rely on small sample sizes or subgroup analyses [41,72], limiting statistical power, and effect sizes are often modest or lose significance after correction for multiple comparisons [67]. Finally, although digital voice markers show promise, their added value over established neuropsychological assessments remains uncertain in several studies [15,41], underscoring the need for longitudinal validation, harmonized protocols, and real-world clinical integration. Collectively, these limitations underscore the need for larger, multilingual cohorts, ecologically valid speech tasks, advanced transcription tools, and models aligned with neuropsychological theory.

## 5. Conclusions and Future Directions

Spontaneous speech analysis, through acoustic and temporal parameters such as silence duration, phrasal segment length, speech segment frequency, and long pauses, offers a rich window into the subtle cognitive and linguistic changes that may signal early memory decline in healthy older adults. These markers, reflecting the natural flow of unscripted verbal communication, are increasingly recognized for their potential in detecting subjective memory complaints and early stages of neurodegenerative disorders. Interdisciplinary research at the intersection of linguistics, neuroscience, and artificial intelligence (AI) has demonstrated that automated tools—especially those based on natural language processing and machine learning—can effectively capture nuanced speech patterns associated with early cognitive changes. These changes align with structural and functional brain alterations in memory- and language-related regions such as the hippocampus and prefrontal cortex. Future directions include integrating real-time speech analytics with neuroimaging and digital biomarkers to enhance early screening and monitoring, as well as implementing multilingual, cross-cultural studies to account for language variability. Such efforts could provide diverse datasets to train generative AI classification pipelines, leading to the development of new speech-based tasks and metrics tailored to detect early cognitive decline across populations. These directions hold promise for transforming spontaneous speech into a scalable, low-cost, and non-invasive diagnostic tool in proactive cognitive health.

## Figures and Tables

**Table 1 healthcare-13-02888-t001:** Review of selected studies according to language indicators in SCD.

Study	Methodology	Τools	Outcomes Measured	Main Findings
Young, 2024 [64]	Participants: 238 cognitively unimpaired adults aged 32–75 from the Framingham Heart Study - Speech analysis: 5 speech markers extracted from delayed recall of a story memory task - Neuroimaging: Amyloid and tau PET, structural MRI - Data processing: transcription, utterance definition, PET and MRI processing	(1) Audio recordings and manual transcription of speech data using the ELAN tool and CHAT format (2) PET imaging for amyloid (PiB) and tau (Flortaucipir) with FreeSurfer processing (3) Structural MRI with FreeSurfer processing	(1) Five speech markers during delayed recall of a story memory task: total utterance time, number of between-utterance pauses, speech rate, and percentage of unique words (2) The associations between these speech markers and global amyloid status and regional tau signal	The study indicates the following: - Speech patterns during delayed recall, including longer between-utterance pause time, more between-utterance pauses, and slower speech rate, were associated with increased tau PET signal across medial temporal and early neocortical regions in cognitively unimpaired adults - These speech markers were not significantly associated with amyloid status or delayed recall scores, suggesting that they capture unique information about early Alzheimer’s disease pathology - The findings indicate that subtle changes in speech patterns during memory recall may reflect early cognitive impairment associated with tau pathology, even before overt clinical symptoms appear
Xu, 2025 [65]	Participants: Use of data from the WRAP and W-ADRC databases (speech recording and cognitive assessments) - Determination of participants’ cognitive status through regular neurological and neuropsychological testing - Division of cognitively unimpaired participants into CU, Aβ(-) and CU, and Aβ(+) groups based on amyloid PET scan results	(1) Praat software to measure fraction of locally unvoiced frames and degree of voice breaks (2) pyannote library to detect unfilled pauses (3) WhisperX ASR tool to detect filled pauses (4) PLLR (phoneme log-likelihood ratio) computed using a DNN-HMM framework to measure articulatory precision	(1) Articulatory precision, as measured by the phoneme log-likelihood ratio (PLLR) (2) Speech fluency, as measured by the fraction of locally unvoiced frames and the degree of voice breaks (3) Speech pace, as measured by word-per-minute rates, information rate, and articulation rate	The study indicates the following: - A reduction in speech fluency and pace was observed among participants with MCI and dementia compared to cognitively unimpaired participants with low amyloid-β burden - The phoneme log-likelihood ratio (PLLR), a measure of articulatory precision, was able to effectively distinguish between cognitively unimpaired participants with low amyloid-β burden and those with MCI or dementia - The PLLR was particularly sensitive in detecting differences between groups with relatively milder cognitive declines, suggesting that alterations in articulatory precision may be an early marker of cognitive decline
Verfaillie, 2019 [40]	Participants: 63 individuals with SCD from the ongoing Subjective Cognitive ImpairmeNt Cohort (SCIENCe) study - Amyloid status was determined using either PET scans or CSF Aβ1-42 levels - Regression to investigate the association between amyloid status and linguistic parameters, adjusting for age, sex, and education	(1) Spontaneous speech recordings using three open-ended questions (2) Verbatim transcription of the speech recordings using PRAAT software (3) Extraction of linguistic parameters from the transcripts using the T-Scan computational linguistics software package (4) Determination of amyloid status using either amyloid PET scans or CSF Aβ1-42 levels	The primary outcomes measured in this study were the associations between amyloid burden and various linguistic parameters derived from spontaneous speech, including specific words (content words, concrete nouns, abstract nouns, conversation fillers), lexical complexity (lemma frequency, Type–Token –Ratio), and syntactic complexity (Developmental Level scale)	The study indicates the following: - High amyloid burden was modestly associated with fewer specific words, particularly in individuals with higher levels of education, but not with lexical or syntactic complexity or performance on conventional neuropsychological language tests - Spontaneous speech recordings may be a more sensitive way to detect subtle language changes related to Alzheimer’s pathology compared to standard neuropsychological assessments
van den Berg, 2025 [66]	Participants: 490 individuals with SCD from two cohorts - Diagnosis of SCD based on typical neurological and neuropsychological assessments - Annual follow-up visits over a mean of 4.3 years, with repeated assessments and reevaluation of diagnoses	(1) Semantic and phonemic verbal fluency tasks (2) CSF Aβ42 levels and amyloid PET scans to measure amyloid burden (3) Annual neurological and neuropsychological assessments to determine clinical progression from SCD to MCI or dementia	(1) Semantic fluency score (2) Phonemic fluency score (3) semantic–phonemic discrepancy score	The study indicates the following: - Amyloid-positive individuals showed faster decline in semantic fluency compared to amyloid-negative individuals, but no difference in phonemic fluency - Decline in both semantic and phonemic fluency was associated with an increased risk of clinical progression to MCI or dementia. - Decline in semantic fluency was an early indicator of cognitive deficits in preclinical Alzheimer’s disease
van den Berg, 2024 [67]	Participants: ≥50 years old, unimpaired cognition, native Dutch speakers, experience with smartphones/tablets - Measurement of Aβ biomarkers-within 1.5 years using PET imaging or CSF analysis, with participants classified as Aβ-positive or Aβ-negative	(1) Use of the Winterlight Assessment app to collect speech samples remotely from participants, including structured (verbal fluency) and unstructured (picture description, journaling) tasks (2) Extraction of over 200 acoustic features from the speech recordings using automatic speech recognition, with 11 features selected a priori based on prior relevance to Alzheimer’s disease, including measures of pauses, phonation, fundamental frequency, and intensity	(1) The feasibility of a remote multi-day tablet-based speech assessment to obtain speech recordings (2) The test–retest reliability of remotely measured acoustic speech features over multiple assessments (3) The associations between remotely measured acoustic speech features and Aβ pathology	The study indicates that - The remote speech assessment was feasible and reliable, with high adherence rates and usability scores - Individuals with positive amyloid-beta (Aβ) status showed higher pause-to-word ratios in picture description and journal-prompt storytelling tasks compared to Aβ-negative individuals, although this difference did not remain significant after correction for multiple testing - The study supports the potential of remotely measured speech acoustics as a promising indicator of subtle cognitive deficits in early Alzheimer’s disease stages.
Soroski, 2022 [68]	- Participants: 72 Individuals (M = 68.8) were recruited from a memory clinic and the community, including those diagnosed with AD, MCI, SMC, and healthy controls.	(1) Three speech tasks to be completed (picture description, reading, and experience recall), which were recorded (1) Google Cloud speech-to-text (STT) used to automatically transcribe speech data (2) Manual correction of the automatic transcripts by human transcribers, including fixing errors, adding punctuation, and annotating filled and silent pauses (3) Computational metrics like word error rate (WER) and match error rate (MER) to evaluate the accuracy of the automatic and manually corrected transcripts	1. Transcription confidence scores 2. Transcription error rates (word error rate and match error rate) 3. Classification accuracy of machine learning models (logistic regression, Gaussian naive Bayes, and random forests) in distinguishing individuals with AD, MCI, or SMC from healthy controls	The study indicates that - The automatic transcription software had higher confidence and lower error rates when transcribing speech from healthy controls compared to patients with Alzheimer’s disease, MCI, or SMC - Manually correcting the transcripts, especially for spontaneous speech tasks, led to significantly improved performance of machine learning models in distinguishing patients from controls, but manual correction did not significantly impact model performance for the reading task - Manually adding pauses to the transcripts did not significantly improve the performance of the machine learning models.
Reeves, 2023 [41]	Participants: 52 individuals with normal cognition to mild dementia performed the Narrative Description (ND) test - Cognitive function was assessed for up to 5 years	(1) The Narrative Description (ND) test and various linguistic features extracted from the transcribed Narrative Descriptions, including word counts, speech rate, lexical diversity, and part-of-speech counts.	The primary outcome measured in this study was the ability to perceive, understand, and describe visual scenes, as assessed by the Narrative Description (ND) test.	The study indicates that - Many linguistic features were related to cognitive status and could distinguish between normal cognition, mild cognitive impairment, and dementia groups - Adding linguistic features to the Narrative Description test did not significantly improve its ability to detect early cognitive impairment or predict future cognitive decline compared to the ND test score alone
König, 2024 [69]	Participants: 78 cases from the PROSPECT-AD study, including cognitively normal individuals and those with SCD, MCI, and dementia	(1) Conducted automated phone-based assessments using the Mili software, which recorded participants’ responses to various tasks including the semantic verbal fluency task (2) Transcribed the semantic verbal fluency task both manually by human raters and automatically using the SIGMA speech analysis pipeline (3) Extracted various speech features from the transcripts, including word count, semantic cluster size and switches, and word frequencies	The primary outcomes measured in this study were the agreement between automated and manual transcriptions of a semantic verbal fluency task, as well as the ability of speech features extracted from these transcriptions to discriminate between cognitively impaired and unimpaired individuals	The study indicates that - Automated speech recognition had high agreement with manual transcription for the quantitative measure of word count, with a 93% probability that the difference would be below the minimally important difference - Qualitative features like semantic cluster size and word frequency had only fair levels of agreement between automated and manual transcription - The word count feature was able to discriminate well between cognitively impaired and unimpaired individuals regardless of the transcription method used.
Konig, 2018 [70]	Participants: 165 through a memory clinic, with SCI, MCI, AD, and mixed dementia recorded while performing vocal cognitive tasks using a mobile application	(1) A mobile application that presented participants with 6 vocal tasks was used (2) A wearable microphone was used to record the participants’ speech (vocal recordings) (3) Speech analysis software and custom signal processing tools were used to extract vocal features from the recorded speech	The primary outcomes measured in this study were (1) To extract vocal markers using speech signal processing and then to test the ability of these markers to distinguish between the different participant groups (SCI, MCI, AD, and mixed dementia) (2) Automatic classifiers were trained using machine learning methods to detect MCI and AD based on the vocal markers	The study demonstrated high classification accuracy in differentiating between SCI, MCI, AD, and mixed dementia (MD) using automatic speech analysis. - Certain vocal tasks, such as the counting down task and the verbal fluency tasks, were particularly useful for distinguishing between the different cognitive impairment groups. - The study suggests that automatic speech analysis using a mobile application can provide clinicians with meaningful information for the assessment and monitoring of people with MCI and AD.
Ter Huurne, 2023 [71]	Participants: 94 individuals from a memory clinic (MCI and SCD) The study used a semiautomated phone assessment with those participants	(1) A semiautomated phone assessment conducted using a mobile application, with a test leader guiding the participant through the tasks (2) The 15-item Verbal Learning Test (VLT) and 1-minute Semantic Verbal Fluency (SVF) task, which were administered as part of the semiautomated phone assessment (3) Manual scoring of the VLT and SVF by the test leader (4) Automatic scoring and extraction of speech and linguistic features from the VLT and SVF using a mobile application with automatic speech recognition (ASR) technology	1. The accuracy of automatic speech recognition (ASR) software in scoring the Verbal Learning Test (VLT) and Semantic Verbal Fluency (SVF) tasks, compared to manual scoring 2. The ability of the VLT and SVF tasks to differentiate between participants with SCD and MCI 3. The additional value of automatically extracted speech and linguistic features from the VLT and SVF tasks in differentiating between SCD and MCI, beyond just the total scores	The study indicates that - The automatic speech recognition (ASR) total word count of the verbal learning test (VLT) and semantic verbal fluency (SVF) tasks were highly comparable to the manually retrieved total word count - The automatically derived speech and linguistic features for the VLT immediate recall and delayed recall had a high diagnostic discriminative power between subjective cognitive decline (SCD) and mild cognitive impairment (MCI) participants - The diagnostic differentiation between SCD and MCI was poor for the SVF task
Hajjar, 2023 [72]	Participants: Collected connected speech, neuropsychological, neuroimaging, and cerebrospinal fluid (CSF) Alzheimer’s disease (AD) biomarker data from 92 cognitively unimpaired (40 Aβ+) and 114 impaired (63 Aβ+) participants	1. Acoustic–semantic features derived from audio recordings using the Geneva Minimalistic Acoustic Parameter Set (GeMAPS) 2. Lexical–semantic features derived from audio recordings using natural language processing (NLP) 3. Structural MRI data, including hippocampal volume and other volumetric measurements 4. Cerebrospinal fluid (CSF) biomarkers, specifically amyloid-beta (Aβ) status	(1) Development of lexical–semantic and acoustic digital voice biomarkers for Alzheimer’s disease (2) Diagnostic performance of the lexical–semantic and acoustic digital voice biomarkers in detecting MCI and amyloid-β status (3) Associations between the digital voice biomarkers and hippocampal volume, CSF amyloid-β, and 2-year disease progression	The study indicates that - Lexical-semantic and acoustic digital voice biomarkers demonstrated higher diagnostic performance for detecting MCI compared to traditional neuropsychological tests - Lexical–semantic digital voice biomarkers were able to detect amyloid-beta (Aβ) status, while acoustic biomarkers were associated with hippocampal volume - Both lexical–semantic and acoustic digital voice biomarkers were associated with 2-year disease progression as measured by changes in the Clinical Dementia Rating-Sum of Boxes (CDR-SOB) score
García-Gutiérrez, 2024 [17]	Participants: 1500 individuals with SCC through a Memory Clinic at the Ace Alzheimer Center, Barcelona - Comprehensive neurological, neuropsychological, and social assessments, including standardized cognitive tests and a final diagnosis by a multidisciplinary team	(1) Task: Describing a picture (the Cookie Theft picture) and naming as many animals as possible in one minute, with the participants’ voices recorded during these tasks (2) Pre-processing of the audio recordings, including standardizing the sampling rate, removing silent portions, and applying noise reduction (3) Extraction of 176 paralinguistic features from the audio data using the eGeMAPS feature set and the OpenSMILE toolkit	The primary outcomes were (1) The ability to differentiate between different clinical phenotypes across the Alzheimer’s disease spectrum, including SCC, MCI, and AD (2) The ability to predict performance in different cognitive domains, including attention, executive functions, language, memory, and visuospatial functions, based on spontaneous speech data	The study indicates that - Machine learning models using paralinguistic features from spontaneous speech were able to identify individuals with AD and MCI - The models were able to predict performance in key cognitive domains like attention, memory, executive function, language, and visuospatial ability with correlations greater than 0.5 - Spontaneous speech analysis has potential for developing screening tools and remote monitoring of cognitive decline in Alzheimer’s disease

## Data Availability

No new data were created or analyzed in this study. Data sharing is not applicable to this article.

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
