# Peer review of "Linguistic Markers in Spontaneous Speech: Insights into Subjective Cognitive Decline (Review)"

_healthcare, 2025, doi:10.3390/healthcare13222888_

Round 1
Reviewer 1 Report
Comments and Suggestions for Authors
Dear Authors,
I have read your manuscript entitled “Linguistic Markers in Spontaneous Speech, Insights into Subjective Cognitive Decline” submitted to Healthcare. The review highlights the potential of spontaneous and connected speech as early markers of cognitive decline and provides a timely synthesis of recent evidence.
While the manuscript is promising and contains many strengths, several revisions are needed to improve clarity, rigor, and accessibility for an international readership.
Major Comments
-
Structure and Focus
-
The manuscript is comprehensive but at times reads as overly descriptive. Consider re-structuring the Results and Discussion to more clearly separate (a) articulatory/prosodic markers, (b) lexical-semantic features, and (c) automated/remote assessment tools. This will help readers identify key themes more easily.
-
Some sections contain overlapping content (e.g., advantages of automated speech recognition appear in both Results and Discussion). Streamlining would improve readability.
-
-
Methodological Considerations
-
Although you state that the review followed PRISMA guidelines, the search strategy and study selection process need greater detail. Please expand the Methods section with the exact search terms, time limits, and number of records retrieved/excluded at each stage. Including a PRISMA flow diagram would be highly beneficial.
-
The criteria distinguishing SMD, SCD, and MCI populations could be explained more consistently, as different studies adopt varying definitions. Clarifying this in the Methods and Limitations would strengthen the review.
-
-
Tables and Figures
-
Table 1 is rich in detail but complex to read in its current form. Consider reorganizing it into sub-sections (e.g., “Speech Fluency,” “Lexical-Semantic,” “Acoustic/Prosodic”) or using multiple smaller tables.
-
A schematic figure summarizing the pathway from subjective memory complaints → spontaneous speech changes → biomarker correlations → clinical outcomes would add significant value. Other schematic explanations could be incorporated into the paper accordingly.
-
-
Critical Appraisal
-
While the review summarizes findings well, it could engage more critically with study limitations. For example, many studies had small samples, relied on different speech tasks, or used heterogeneous biomarker definitions. Highlighting these methodological differences explicitly will give readers a clearer understanding of the evidence base.
-
Similarly, the discussion could better balance enthusiasm for digital biomarkers with caution about their current limitations (e.g., reliability of ASR systems, lack of cross-linguistic validation).
-
-
Language and Style
-
The manuscript would benefit from careful language editing to reduce repetition and improve clarity. For instance, “spontaneous speech offers a rich window into subtle cognitive changes” is stated multiple times.
-
Acronyms (SMD, SCD, SCC, ASR, NLP) should be defined at first mention and used consistently throughout.
-
Minor Comments
-
Please ensure consistent reference formatting according to Healthcare journal guidelines.
-
The Conclusions could be shortened and made more focused, highlighting 2–3 practical take-home messages for clinicians and researchers.
-
A “Future Directions” subsection could be expanded to outline how large-scale, multilingual, and longitudinal studies may help validate these speech-based markers.
This is a valuable and timely review that has the potential to contribute to the field significantly. With revisions to strengthen the methodology description, improve structure, and sharpen the critical discussion, the manuscript will be much improved.
I look forward to reading a revised version.
With kind regards,
Comments on the Quality of English LanguageEnglish is well written. However, to increase fluency and reduce repetition, it would be valuable for a professional editing service.
Author Response
Reviewer 1
|
Comment |
|
|
-Thank you for your comment. A more readable and comprehensive structure is now given. The document now follows a more consistent structure regarding the results and discussion, answering also to the iniitial research questions. The structure now in the results is: Articulatory and Prosodic Features, Lexical-Semantic and Language Content Features, Verbal Fluency features, tools and methods to analyse spontaneous and connected speech -indeed, we have now revised the manuscript according the overlapping information. We keep part of advantages of automated speech in discussion part instead of results part. |
|
-Indeed, although this is not a systematic but a literature review, we follow the Prisma guidelines, and we now have added the Prisma flow chart. -In methods, we added also the exact search terms (not in a supplementary material but in the text), and number of records retrieved /excluded at each stage giving the reason. The Prisma flow chart has all the details.
- Subjective memory is a construct to be quantified, studied, and understood in clinical practice. It can be referred to as subjective memory complaints (SMC), subjective memory decline (SMD), subjective cognitive decline (SCD), subjective memory impairment (SMI), or subjective cognitive impairment (SCI) (Edmonds et al. 2014; Stewart 2012) as we mentioned in the intriduction. The definitions vary across the literature. The SCD-I working group notes that SCD has been described under multiple terms, including SMI and SCC, and recommends using the term “cognitive” rather than “memory,” since early decline may not be limited to memory (Jesen et al., 2014). We have added in limitation these differentiate in definition. Although, subjective cognitive decline / memory complaints have specific criteria given by SCD-I working group (see introduction section), they are not by themselves a diagnosable disorder in ICD-11. ICD-11 has diagnostic categories for Mild Neurocognitive Disorder and Major Neurocognitive Disorder (roughly analogous to MCI and dementia).
-MCI have specific criteria according to Petersen et al (1999). The definition and the criteria are given in the introduction. |
|
-In Table 1, we added a column titled “AIM” according to the main goal o studies. We categorized the studies as Verbal Fluency (VF), Lexical-Semantic (L/S), or Acoustic/Prosodic (A/P). Results from some studies were categorized under more than one type/aim. - Thank you for this suggestion. While we agree that a schematic illustration could indeed be useful, at this stage it would be premature to depict a single, linear pathway, as the evidence does not yet clearly support such a model. Instead, the current body of literature highlights heterogeneous findings and methodological variability, suggesting that more research is needed before a definitive pathway can be established. The main aim of this review is therefore to provide a clearer overview of the linguistic features that may change in the context of subjective cognitive decline and to examine the methods currently being used to study these changes, in order to inform and refine future research questions. We believe that this approach avoids oversimplification while remaining faithful to the state of the evidence.
|
|
-Thank you for this comment. The section of limitation in the last paragraph of discussion has now revised. We have added the small samples, heterogeneity in speech tasks, heterogenous SMD definition , and methodological heterogenity.
-We have also added more info about the limitation of ASR systems, and the lack of cross-linguistic validation |
|
-Indeed , thank you. We now revised the repetitions - All acronyms are now defined at their first mention. They are also listed in detail at the end of the manuscript. |
References
Gough, 2017: https://uk.sagepub.com/sites/default/files/upm-assets/81596_book_item_81596.pdf
Reviewer 2 Report
Comments and Suggestions for Authors
Refer to the attached file for further details.

Author Response
Reviewer 2.
|
Comment
|
Answer |
|
‘’This paper examines spontaneous speech in individuals with subjective memory decline (SMD), showing that reduced fluency, slower speech, and increased pauses may serve as early markers of Alzheimer’s disease, highlighting the clinical potential of speech analysis. Overall, this manuscript makes a meaningful contribution to the field of cognitive decline assessment, particularly by highlighting spontaneous speech analysis as a standardized and non-invasive approach for the early detection of subjective memory decline (SMD) and related Alzheimer’s disease pathology. Nevertheless, several improvements are recommended to strengthen the clarity, methodological transparency, and clinical applicability of the work’’
|
We would like to thank the reviewer for her/his valuable time and feedback. Following her/his comments, the document has been substantially revised. |
|
1. Alignment between Research Questions and Conclusions The manuscript clearly outlines three research questions; however, the Discussion and Conclusions sections do not explicitly provide structured answers to each. I strongly recommend that the authors present subconclusions corresponding to each research question. This would enhance the coherence of the paper and strengthen the overall persuasiveness of the findings. |
Thank you for your comment. Actually, the research questions are answered by the subsection of the results and discussion but of course we understand that it can be clearer to the reader. Specifically, research questions are: 1 What linguistic features are used to detect cognitive decline in spontaneous speech of individuals with SCD? 2.What tools or methods are employed to analyze spontaneous and connected speech. We have now revised the subsections of results in order to directly answer to research questions : Articulatory and Prosodic Features, Lexical-Semantic and Language Content Features, Verbal Fluency features, tools and methods to analyse spontaneous and connected speech
|
|
2. Methodological Detail and PRISMA Compliance The search strategy currently relies only on PubMed and Cochrane. To ensure comprehensiveness, the inclusion of additional databases such as Web of Science, Scopus, or PsycINFO should be considered. If these were intentionally excluded, the rationale should be explicitly stated. Moreover, the manuscript refers to adherence to PRISMA guidelines but does not include a PRISMA flow diagram. This figure is essential for transparency and should be incorporated. |
-Extraction of the studies conducted only from two data bases (cohrane and Pubmed) as described in the methods. Since this is not a systematic review, but rather a narrative literature review, the use of two major, high-quality databases (PubMed and Cochrane Library) is methodologically acceptable. The purpose of a narrative review is not to achieve exhaustive coverage, but to provide a focused and critical synthesis of the most relevant evidence. In narrative reviews, it is common practice to limit the number of databases for reasons of relevance, feasibility, and focus (Gough et al., 2017, An introduction to systematic reviews). The key is transparency in describing the search strategy rather than exhaustiveness. However, we added this info as limitation (please see the limitation section). -we now added the PRISMA flow chart
|
|
3. Interpretation of Table 1 While Table 1 provides a useful summary of the included studies, the accompanying discussion is repetitive and does not sufficiently highlight the strengths and limitations of each study. I recommend going beyond summary to offer a more comparative and interpretative synthesis, akin to a meta-analytic discussion, to enhance the depth of the results section. |
Thank you for this comment, we have now revised the discussion part with a more critical insight. |
|
4. Terminology Consistency The manuscript alternates between the terms SMD, SCD, SMI, and SCC, which introduces ambiguity. These terms should be clearly defined at the beginning of the manuscript, and consistent terminology should be maintained throughout. |
Thank you for your comment, indeed there is a frustration of definitions. However, we explain in the introduction that all the terms describe the same situation. Specifically, Subjective memory is a construct to be quantified, studied, and understood in clinical practice. It can be referred to as subjective memory complaints (SMC), subjective memory decline (SMD), subjective cognitive decline (SCD), subjective memory impairment (SMI), or subjective cognitive impairment (SCI) (Edmonds et al. 2014; Stewart 2012). The definitions vary across the literature. The SCD-I working group notes that SCD has been described under multiple terms, including SMI and SCC, and recommends using the term “cognitive” rather than “memory,” since early decline may not be limited to memory. We have added in limitation these differentiate in definition. Although, subjective cognitive decline / memory complaints have specific criteria given by SCD-I working group (see introduction section), they are not by themselves a diagnosable disorder in ICD-11. The abbreviations of definitions are places at the end of manuscript
|
|
5. Limitations Although limitations are acknowledged, the discussion remains overly general. A more precise articulation of limitations would improve the manuscript. In particular, I suggest emphasizing: • methodological heterogeneity across studies, • small and localized sample sizes, and • reliability concerns regarding automatic speech recognition (ASR) outputs. |
We have now revised the separate paragraph for limitations at the end of discussion. Thank you for your invaluable comment. We agree and we added the suggested limitation. |
|
6. Future Directions The Future Directions section appropriately highlights the need for multilingual and cross-cultural studies. However, this discussion would be more impactful if it extended to concrete applications, such as the integration of speech biomarkers into clinical workflows, their incorporation into digital healthcare platforms for remote monitoring, and their scalability for early screening in at risk populations. |
Thank you for this invaluable comment. We have now incorporate this in the future direction section. |
References
Gough, 2017: https://uk.sagepub.com/sites/default/files/upm-assets/81596_book_item_81596.pdf
Round 2
Reviewer 1 Report
Comments and Suggestions for Authors
Dear Authors,
I have carefully reviewed the revised version of your manuscript entitled “Linguistic Markers in Spontaneous Speech, Insights into Subjective Cognitive Decline.” I appreciate the effort you have invested in addressing the previous concerns.
The restructuring of the Results and Discussion into clearer thematic categories has greatly improved readability, and the removal of overlapping content has enhanced focus. The inclusion of the detailed search strategy and PRISMA flow chart strengthens methodological transparency. Clarifications on the definitions of SMD, SCD, and MCI populations are well integrated, and the expanded limitations section now offers a balanced critical appraisal of methodological variability and digital biomarker constraints.
Table 1 is more accessible with the new categorization, and although you chose not to include a schematic figure, your rationale is reasonable and scientifically sound. The manuscript’s language has also been substantially improved, with reduced repetition and consistent use of acronyms.
Overall, this revised version represents a significant improvement and provides a valuable synthesis of the current state of research in this emerging field. I find the work suitable for publication in Healthcare and would like to recommend that you accept it.
Sincerely.
Reviewer 2 Report
Comments and Suggestions for Authors
All reviewer comments have been addressed in this revised manuscript.